# Variational Dropout and the Local Reparameterization Trick

**Diederik P. Kingma**$^*$**, Tim Salimans**$^\times$ **and Max Welling**$^{*\dagger}$
$^*$ Machine Learning Group, University of Amsterdam
$^\times$ Algoritmica
$^\dagger$ University of California, Irvine, and the Canadian Institute for Advanced Research (CIFAR)
`D.P.Kingma@uva.nl, salimans.tim@gmail.com, M.Welling@uva.nl`

## Abstract

We investigate a local reparameterizaton technique for greatly reducing the variance of stochastic gradients for variational Bayesian inference (SGVB) of a posterior over model parameters, while retaining parallelizability. This local reparameterization translates uncertainty about global parameters into local noise that is independent across datapoints in the minibatch. Such parameterizations can be trivially parallelized and have variance that is inversely proportional to the minibatch size, generally leading to much faster convergence. Additionally, we explore a connection with dropout: Gaussian dropout objectives correspond to SGVB with local reparameterization, a scale-invariant prior and proportionally fixed posterior variance. Our method allows inference of more flexibly parameterized posteriors; specifically, we propose *variational dropout*, a generalization of Gaussian dropout where the dropout rates are learned, often leading to better models. The method is demonstrated through several experiments.

## 1 Introduction

Deep neural networks are a flexible family of models that easily scale to millions of parameters and datapoints, but are still tractable to optimize using minibatch-based stochastic gradient ascent. Due to their high flexibility, neural networks have the capacity to fit a wide diversity of nonlinear patterns in the data. This flexbility often leads to *overfitting* when left unchecked: spurious patterns are found that happen to fit well to the training data, but are not predictive for new data. Various regularization techniques for controlling this overfitting are used in practice; a currently popular and empirically effective technique being *dropout* [10]. In [22] it was shown that regular (binary) dropout has a Gaussian approximation called *Gaussian dropout* with virtually identical regularization performance but much faster convergence. In section 5 of [22] it is shown that Gaussian dropout optimizes a lower bound on the marginal likelihood of the data. In this paper we show that a relationship between dropout and Bayesian inference can be extended and exploited to greatly improve the efficiency of variational Bayesian inference on the model parameters. This work has a direct interpretation as a generalization of Gaussian dropout, with the same fast convergence but now with the freedom to specify more flexibly parameterized posterior distributions.

Bayesian posterior inference over the neural network parameters is a theoretically attractive method for controlling overfitting; exact inference is computationally intractable, but efficient approximate schemes can be designed. Markov Chain Monte Carlo (MCMC) is a class of approximate inference methods with asymptotic guarantees, pioneered by [16] for the application of regularizing neural networks. Later useful refinements include [23] and [1].

An alternative to MCMC is variational inference [11] or the equivalent *minimum description length* (MDL) framework. Modern variants of stochastic variational inference have been applied to neural

networks with some succes [8], but have been limited by high variance in the gradients. Despite their theoretical attractiveness, Bayesian methods for inferring a posterior distribution over neural network weights have not yet been shown to outperform simpler methods such as dropout. Even a new crop of efficient variational inference algorithms based on stochastic gradients with minibatches of data [14, 17, 19] have not yet been shown to significantly improve upon simpler dropout-based regularization.

In section 2 we explore an as yet unexploited trick for improving the efficiency of stochastic gradient-based variational inference with minibatches of data, by translating uncertainty about global parameters into local noise that is independent across datapoints in the minibatch. The resulting method has an optimization speed on the same level as fast dropout [22], and indeed has the original Gaussian dropout method as a special case. An advantage of our method is that it allows for full Bayesian analysis of the model, and that it's significantly more flexible than standard dropout. The approach presented here is closely related to several popular methods in the literature that regularize by adding random noise; these relationships are discussed in section 4.

## 2   Efficient and Practical Bayesian Inference

We consider Bayesian analysis of a dataset $\mathcal{D}$, containing a set of $N$ i.i.d. observations of tuples $(\mathbf{x}, \mathbf{y})$, where the goal is to learn a model with *parameters* or *weights* $\mathbf{w}$ of the conditional probability $p(\mathbf{y}|\mathbf{x}, \mathbf{w})$ (standard classification or regression)[1]. Bayesian inference in such a model consists of updating some initial belief over parameters $\mathbf{w}$ in the form of a prior distribution $p(\mathbf{w})$, after observing data $\mathcal{D}$, into an updated belief over these parameters in the form of (an approximation to) the posterior distribution $p(\mathbf{w}|\mathcal{D})$. Computing the true posterior distribution through Bayes' rule $p(\mathbf{w}|\mathcal{D}) = p(\mathbf{w})p(\mathcal{D}|\mathbf{w})/p(\mathcal{D})$ involves computationally intractable integrals, so good approximations are necessary. In *variational inference*, inference is cast as an optimization problem where we optimize the parameters $\phi$ of some parameterized model $q_\phi(\mathbf{w})$ such that $q_\phi(\mathbf{w})$ is a close approximation to $p(\mathbf{w}|\mathcal{D})$ as measured by the Kullback-Leibler divergence $D_{KL}(q_\phi(\mathbf{w})||p(\mathbf{w}|\mathcal{D}))$. This divergence of our posterior $q_\phi(\mathbf{w})$ to the true posterior is minimized in practice by maximizing the so-called variational lower bound $\mathcal{L}(\phi)$ of the marginal likelihood of the data:

$$\mathcal{L}(\phi) = -D_{KL}(q_\phi(\mathbf{w})||p(\mathbf{w})) + L_\mathcal{D}(\phi) \tag{1}$$

$$\text{where} \quad L_\mathcal{D}(\phi) = \sum_{(\mathbf{x},\mathbf{y})\in\mathcal{D}} \mathbb{E}_{q_\phi(\mathbf{w})} \left[\log p(\mathbf{y}|\mathbf{x}, \mathbf{w})\right] \tag{2}$$

We'll call $L_\mathcal{D}(\phi)$ the *expected log-likelihood*. The bound $\mathcal{L}(\phi)$ plus $D_{KL}(q_\phi(\mathbf{w})||p(\mathbf{w}|\mathcal{D}))$ equals the (conditional) marginal log-likelihood $\sum_{(\mathbf{x},\mathbf{y})\in\mathcal{D}} \log p(\mathbf{y}|\mathbf{x})$. Since this marginal log-likelihood is constant w.r.t. $\phi$, maximizing the bound w.r.t. $\phi$ will minimize $D_{KL}(q_\phi(\mathbf{w})||p(\mathbf{w}|\mathcal{D}))$.

### 2.1   Stochastic Gradient Variational Bayes (SGVB)

Various algorithms for gradient-based optimization of the variational bound (eq. (1)) with differentiable $q$ and $p$ exist. See section 4 for an overview. A recently proposed efficient method for minibatch-based optimization with differentiable models is the *stochastic gradient variational Bayes* (SGVB) method introduced in [14] (especially appendix F) and [17]. The basic trick in SGVB is to parameterize the random parameters $\mathbf{w} \sim q_\phi(\mathbf{w})$ as: $\mathbf{w} = f(\epsilon, \phi)$ where $f(.)$ is a differentiable function and $\epsilon \sim p(\epsilon)$ is a random noise variable. In this new parameterisation, an unbiased differentiable minibatch-based Monte Carlo estimator of the expected log-likelihood can be formed:

$$L_\mathcal{D}(\phi) \simeq L_\mathcal{D}^{\text{SGVB}}(\phi) = \frac{N}{M} \sum_{i=1}^{M} \log p(\mathbf{y}^i|\mathbf{x}^i, \mathbf{w} = f(\epsilon, \phi)), \tag{3}$$

where $(\mathbf{x}^i, \mathbf{y}^i)_{i=1}^{M}$ is a minibatch of data with $M$ random datapoints $(\mathbf{x}^i, \mathbf{y}^i) \sim \mathcal{D}$, and $\epsilon$ is a noise vector drawn from the noise distribution $p(\epsilon)$. We'll assume that the remaining term in the variational lower bound, $D_{KL}(q_\phi(\mathbf{w})||p(\mathbf{w}))$, can be computed deterministically, but otherwise it may be approximated similarly. The estimator (3) is differentiable w.r.t. $\phi$ and unbiased, so its gradient

is also unbiased: $\nabla_\phi L_\mathcal{D}(\phi) \simeq \nabla_\phi L_\mathcal{D}^{\text{SGVB}}(\phi)$. We can proceed with variational Bayesian inference by randomly initializing $\phi$ and performing stochastic gradient ascent on $\mathcal{L}(\phi)$ (1).

## 2.2 Variance of the SGVB estimator

The theory of stochastic approximation tells us that stochastic gradient ascent using (3) will asymptotically converge to a local optimum for an appropriately declining step size and sufficient weight updates [18], but in practice the performance of stochastic gradient ascent crucially depends on the variance of the gradients. If this variance is too large, stochastic gradient descent will fail to make much progress in any reasonable amount of time. Our objective function consists of an expected log likelihood term that we approximate using Monte Carlo, and a KL divergence term $D_{KL}(q_\phi(\mathbf{w})||p(\mathbf{w}))$ that we assume can be calculated analytically and otherwise be approximated with Monte Carlo with similar reparameterization.

Assume that we draw minibatches of datapoints with replacement; see appendix F for a similar analysis for minibatches without replacement. Using $L_i$ as shorthand for $\log p(\mathbf{y}^i|\mathbf{x}^i, \mathbf{w} = f(\epsilon^i, \phi))$, the contribution to the likelihood for the $i$-th datapoint in the minibatch, the Monte Carlo estimator (3) may be rewritten as $L_\mathcal{D}^{\text{SGVB}}(\phi) = \frac{N}{M}\sum_{i=1}^M L_i$, whose variance is given by

$$\text{Var}\left[L_\mathcal{D}^{\text{SGVB}}(\phi)\right] = \frac{N^2}{M^2}\Big(\sum_{i=1}^M \text{Var}\left[L_i\right] + 2\sum_{i=1}^M \sum_{j=i+1}^M \text{Cov}\left[L_i, L_j\right]\Big) \tag{4}$$

$$= N^2\Big(\frac{1}{M}\text{Var}\left[L_i\right] + \frac{M-1}{M}\text{Cov}\left[L_i, L_j\right]\Big), \tag{5}$$

where the variances and covariances are w.r.t. both the data distribution and $\epsilon$ distribution, i.e. $\text{Var}\left[L_i\right] = \text{Var}_{\epsilon,\mathbf{x}^i,\mathbf{y}^i}\left[\log p(\mathbf{y}^i|\mathbf{x}^i, \mathbf{w} = f(\epsilon, \phi))\right]$, with $\mathbf{x}^i, \mathbf{y}^i$ drawn from the empirical distribution defined by the training set. As can be seen from (5), the total contribution to the variance by $\text{Var}\left[L_i\right]$ is inversely proportional to the minibatch size $M$. However, the total contribution by the covariances does not decrease with $M$. In practice, this means that the variance of $L_\mathcal{D}^{\text{SGVB}}(\phi)$ can be dominated by the covariances for even moderately large $M$.

## 2.3 Local Reparameterization Trick

We therefore propose an alternative estimator for which we have $\text{Cov}\left[L_i, L_j\right] = 0$, so that the variance of our stochastic gradients scales as $1/M$. We then make this new estimator computationally efficient by not sampling $\epsilon$ directly, but only sampling the intermediate variables $f(\epsilon)$ through which $\epsilon$ influences $L_\mathcal{D}^{\text{SGVB}}(\phi)$. By doing so, the global uncertainty in the weights is translated into a form of local uncertainty that is independent across examples and easier to sample. We refer to such a reparameterization from global noise to local noise as the *local reparameterization trick*. Whenever a source of global noise can be translated to local noise in the intermediate states of computation ($\epsilon \rightarrow f(\epsilon)$), a local reparameterization can be applied to yield a computationally and statistically efficient gradient estimator.

Such local reparameterization applies to a fairly large family of models, but is best explained through a simple example: Consider a standard fully connected neural network containing a hidden layer consisting of 1000 neurons. This layer receives an $M \times 1000$ input feature matrix $\mathbf{A}$ from the layer below, which is multiplied by a $1000 \times 1000$ weight matrix $\mathbf{W}$, before a nonlinearity is applied, i.e. $\mathbf{B} = \mathbf{AW}$. We then specify the posterior approximation on the weights to be a fully factorized Gaussian, i.e. $q_\phi(w_{i,j}) = N(\mu_{i,j}, \sigma_{i,j}^2) \forall w_{i,j} \in \mathbf{W}$, which means the weights are sampled as $w_{i,j} = \mu_{i,j} + \sigma_{i,j}\epsilon_{i,j}$, with $\epsilon_{i,j} \sim N(0,1)$. In this case we could make sure that $\text{Cov}\left[L_i, L_j\right] = 0$ by sampling a separate weight matrix $\mathbf{W}$ for each example in the minibatch, but this is not computationally efficient: we would need to sample $M$ million random numbers for just a single layer of the neural network. Even if this could be done efficiently, the computation following this step would become much harder: Where we originally performed a simple matrix-matrix product of the form $\mathbf{B} = \mathbf{AW}$, this now turns into $M$ separate local vector-matrix products. The theoretical complexity of this computation is higher, but, more importantly, such a computation can usually not be performed in parallel using fast device-optimized BLAS (Basic Linear Algebra Subprograms). This also happens with other neural network architectures such as convolutional neural networks, where optimized libraries for convolution cannot deal with separate filter matrices per example.

Fortunately, the weights (and therefore $\epsilon$) only influence the expected log likelihood through the neuron activations $\mathbf{B}$, which are of much lower dimension. If we can therefore sample the random activations $\mathbf{B}$ directly, without sampling $\mathbf{W}$ or $\epsilon$, we may obtain an efficient Monte Carlo estimator at a much lower cost. For a factorized Gaussian posterior on the weights, the posterior for the activations (conditional on the input $\mathbf{A}$) is also factorized Gaussian:

$$q_\phi(w_{i,j}) = N(\mu_{i,j}, \sigma_{i,j}^2) \; \forall w_{i,j} \in \mathbf{W} \quad \Longrightarrow \quad q_\phi(b_{m,j}|\mathbf{A}) = N(\gamma_{m,j}, \delta_{m,j}), \text{ with}$$

$$\gamma_{m,j} = \sum_{i=1}^{1000} a_{m,i}\mu_{i,j}, \quad \text{and} \quad \delta_{m,j} = \sum_{i=1}^{1000} a_{m,i}^2 \sigma_{i,j}^2. \tag{6}$$

Rather than sampling the Gaussian weights and then computing the resulting activations, we may thus sample the activations from their implied Gaussian distribution directly, using $b_{m,j} = \gamma_{m,j} + \sqrt{\delta_{m,j}}\zeta_{m,j}$, with $\zeta_{m,j} \sim N(0,1)$. Here, $\zeta$ is an $M \times 1000$ matrix, so we only need to sample $M$ thousand random variables instead of $M$ million: a thousand fold savings.

In addition to yielding a gradient estimator that is more *computationally efficient* than drawing separate weight matrices for each training example, the local reparameterization trick also leads to an estimator that has *lower variance*. To see why, consider the stochastic gradient estimate with respect to the posterior parameter $\sigma_{i,j}^2$ for a minibatch of size $M = 1$. Drawing random weights $\mathbf{W}$, we get

$$\frac{\partial L_\mathcal{D}^{\text{SGVB}}}{\partial \sigma_{i,j}^2} = \frac{\partial L_\mathcal{D}^{\text{SGVB}}}{\partial b_{m,j}} \frac{\epsilon_{i,j}a_{m,i}}{2\sigma_{i,j}}. \tag{7}$$

If, on the other hand, we form the same gradient using the local reparameterization trick, we get

$$\frac{\partial L_\mathcal{D}^{\text{SGVB}}}{\partial \sigma_{i,j}^2} = \frac{\partial L_\mathcal{D}^{\text{SGVB}}}{\partial b_{m,j}} \frac{\zeta_{m,j}a_{m,i}^2}{2\sqrt{\delta_{m,j}}}. \tag{8}$$

Here, there are two stochastic terms: The first is the backpropagated gradient $\partial L_\mathcal{D}^{\text{SGVB}}/\partial b_{m,j}$, and the second is the sampled random noise ($\epsilon_{i,j}$ or $\zeta_{m,j}$). Estimating the gradient with respect to $\sigma_{i,j}^2$ then basically comes down to estimating the covariance between these two terms. This is much easier to do for $\zeta_{m,j}$ as there are much fewer of these: individually they have higher correlation with the backpropagated gradient $\partial L_\mathcal{D}^{\text{SGVB}}/\partial b_{m,j}$, so the covariance is easier to estimate. In other words, measuring the effect of $\zeta_{m,j}$ on $\partial L_\mathcal{D}^{\text{SGVB}}/\partial b_{m,j}$ is easy as $\zeta_{m,j}$ is the only random variable directly influencing this gradient via $b_{m,j}$. On the other hand, when sampling random weights, there are a thousand $\epsilon_{i,j}$ influencing each gradient term, so their individual effects get lost in the noise. In appendix D we make this argument more rigorous, and in section 5 we show that it holds experimentally.

## 3  Variational Dropout

*Dropout* is a technique for regularization of neural network parameters, which works by adding multiplicative noise to the input of each layer of the neural network during optimization. Using the notation of section 2.3, for a fully connected neural network dropout corresponds to:

$$\mathbf{B} = (\mathbf{A} \circ \xi)\theta, \quad \text{with } \xi_{i,j} \sim p(\xi_{i,j}) \tag{9}$$

where $\mathbf{A}$ is the $M \times K$ matrix of input features for the current minibatch, $\theta$ is a $K \times L$ weight matrix, and $\mathbf{B}$ is the $M \times L$ output matrix for the current layer (before a nonlinearity is applied). The $\circ$ symbol denotes the elementwise (Hadamard) product of the input matrix with a $M \times K$ matrix of independent noise variables $\xi$. By adding noise to the input during training, the weight parameters $\theta$ are less likely to overfit to the training data, as shown empirically by previous publications. Originally, [10] proposed drawing the elements of $\xi$ from a Bernoulli distribution with probability $1 - p$, with $p$ the *dropout rate*. Later it was shown that using a continuous distribution with the same relative mean and variance, such as a Gaussian $N(1, \alpha)$ with $\alpha = p/(1-p)$, works as well or better [20].

Here, we re-interpret dropout with continuous noise as a variational method, and propose a generalization that we call *variational dropout*. In developing variational dropout we provide a firm Bayesian justification for dropout training by deriving its implicit prior distribution and variational objective. This new interpretation allows us to propose several useful extensions to dropout, such as a principled way of making the normally fixed dropout rates $p$ *adaptive* to the data.

## 3.1 Variational dropout with independent weight noise

If the elements of the noise matrix $\xi$ are drawn independently from a Gaussian $N(1, \alpha)$, the marginal distributions of the activations $b_{m,j} \in \mathbf{B}$ are Gaussian as well:

$$q_\phi(b_{m,j}|\mathbf{A}) = N(\gamma_{m,j}, \delta_{m,j}), \text{ with } \gamma_{m,j} = \sum_{i=1}^{K} a_{m,i}\theta_{i,j}, \text{ and } \delta_{m,j} = \alpha \sum_{i=1}^{K} a_{m,i}^2 \theta_{i,j}^2. \quad (10)$$

Making use of this fact, [22] proposed *Gaussian dropout*, a regularization method where, instead of applying (9), the activations are directly drawn from their (approximate or exact) marginal distributions as given by (10). [22] argued that these marginal distributions are exact for Gaussian noise $\xi$, and for Bernoulli noise still approximately Gaussian because of the central limit theorem. This ignores the dependencies between the different elements of $\mathbf{B}$, as present using (9), but [22] report good results nonetheless.

As noted by [22], and explained in appendix B, this Gaussian dropout noise can also be interpreted as arising from a Bayesian treatment of a neural network with weights $\mathbf{W}$ that multiply the input to give $\mathbf{B} = \mathbf{AW}$, where the posterior distribution of the weights is given by a factorized Gaussian with $q_\phi(w_{i,j}) = \mathcal{N}(\theta_{i,j}, \alpha\theta_{i,j}^2)$. From this perspective, the marginal distributions (10) then arise through the application of the local reparameterization trick, as introduced in section 2.3. The variational objective corresponding to this interpretation is discussed in section 3.3.

## 3.2 Variational dropout with correlated weight noise

Instead of ignoring the dependencies of the activation noise, as in section 3.1, we may retain the dependencies by interpreting dropout (9) as a form of *correlated weight noise*:

$$\mathbf{B} = (\mathbf{A} \circ \xi)\theta, \ \xi_{i,j} \sim N(1, \alpha) \iff \mathbf{b}^m = \mathbf{a}^m \mathbf{W}, \text{ with}$$
$$\mathbf{W} = (\mathbf{w}_1', \mathbf{w}_2', \dots, \mathbf{w}_K')', \text{ and } \mathbf{w}_i = s_i \theta_i, \text{ with } q_\phi(s_i) = N(1, \alpha), \quad (11)$$

where $\mathbf{a}^m$ is a row of the input matrix and $\mathbf{b}^m$ a row of the output. The $\mathbf{w}_i$ are the rows of the weight matrix, each of which is constructed by multiplying a non-stochastic parameter vector $\theta_i$ by a stochastic scale variable $s_i$. The distribution on these scale variables we interpret as a Bayesian posterior distribution. The weight parameters $\theta_i$ (and the biases) are estimated using maximum likelihood. The original Gaussian dropout sampling procedure (9) can then be interpreted as arising from a local reparameterization of our posterior on the weights $\mathbf{W}$.

## 3.3 Dropout's scale-invariant prior and variational objective

The posterior distributions $q_\phi(\mathbf{W})$ proposed in sections 3.1 and 3.2 have in common that they can be decomposed into a parameter vector $\theta$ that captures the mean, and a multiplicative noise term determined by parameters $\alpha$. Any posterior distribution on $\mathbf{W}$ for which the noise enters this multiplicative way, we will call a *dropout posterior*. Note that many common distributions, such as univariate Gaussians (with nonzero mean), can be reparameterized to meet this requirement.

During dropout training, $\theta$ is adapted to maximize the expected log likelihood $\mathbb{E}_{q_\alpha}[L_\mathcal{D}(\theta)]$. For this to be consistent with the optimization of a variational lower bound of the form in (2), the prior on the weights $p(\mathbf{w})$ has to be such that $D_{KL}(q_\phi(\mathbf{w})||p(\mathbf{w}))$ does not depend on $\theta$. In appendix C we show that the only prior that meets this requirement is the scale invariant log-uniform prior:

$$p(\log(|w_{i,j}|)) \propto c,$$

i.e. a prior that is uniform on the log-scale of the weights (or the weight-scales $s_i$ for section 3.2). As explained in appendix A, this prior has an interesting connection with the floating point format for storing numbers: From an MDL perspective, the floating point format is optimal for communicating numbers drawn from this prior. Conversely, the KL divergence $D_{KL}(q_\phi(\mathbf{w})||p(\mathbf{w}))$ with this prior has a natural interpretation as regularizing the number of significant digits our posterior $q_\phi$ stores for the weights $w_{i,j}$ in the floating-point format.

Putting the expected log likelihood and KL-divergence penalty together, we see that dropout training maximizes the following variatonal lower bound w.r.t. $\theta$:

$$\mathbb{E}_{q_\alpha}[L_\mathcal{D}(\theta)] - D_{KL}(q_\alpha(\mathbf{w})||p(\mathbf{w})), \quad (12)$$

where we have made the dependence on the $\theta$ and $\alpha$ parameters explicit. The noise parameters $\alpha$ (e.g. the dropout rates) are commonly treated as hyperparameters that are kept fixed during training. For the log-uniform prior this then corresponds to a fixed limit on the number of significant digits we can learn for each of the weights $w_{i,j}$. In section 3.4 we discuss the possibility of making this limit adaptive by also maximizing the lower bound with respect to $\alpha$.

For the choice of a factorized Gaussian approximate posterior with $q_{\phi}(w_{i,j}) = \mathcal{N}(\theta_{i,j}, \alpha\theta_{i,j}^2)$, as discussed in section 3.1, the lower bound (12) is analyzed in detail in appendix C. There, it is shown that for this particular choice of posterior the negative KL-divergence $-D_{KL}(q_{\alpha}(\mathbf{w})||p(\mathbf{w}))$ is not analytically tractable, but can be approximated extremely accurately using

$$-D_{KL}[q_{\phi}(w_i)|p(w_i)] \approx \text{constant} + 0.5\log(\alpha) + c_1\alpha + c_2\alpha^2 + c_3\alpha^3,$$

with

$$c_1 = 1.16145124, \quad c_2 = -1.50204118, \quad c_3 = 0.58629921.$$

The same expression may be used to calculate the corresponding term $-D_{KL}(q_{\alpha}(\mathbf{s})||p(\mathbf{s}))$ for the posterior approximation of section 3.2.

## 3.4 Adaptive regularization through optimizing the dropout rate

The noise parameters $\alpha$ used in dropout training (e.g. the dropout rates) are usually treated as fixed hyperparameters, but now that we have derived dropout's variational objective (12), making these parameters adaptive is trivial: simply maximize the variational lower bound with respect to $\alpha$. We can use this to learn a separate dropout rate per layer, per neuron, of even per separate weight. In section 5 we look at the predictive performance obtained by making $\alpha$ adaptive.

We found that very large values of $\alpha$ correspond to local optima from which it is hard to escape due to large-variance gradients. To avoid such local optima, we found it beneficial to set a constraint $\alpha \leq 1$ during training, i.e. we maximize the posterior variance at the square of the posterior mean, which corresponds to a dropout rate of $0.5$.

## 4 Related Work

Pioneering work in practical variational inference for neural networks was done in [8], where a (biased) variational lower bound estimator was introduced with good results on recurrent neural network models. In later work [14, 17] it was shown that even more practical estimators can be formed for most types of continuous latent variables or parameters using a (non-local) reparameterization trick, leading to efficient and unbiased stochastic gradient-based variational inference. These works focused on an application to latent-variable inference; extensive empirical results on inference of global model parameters were reported in [6], including succesful application to reinforcement learning. These earlier works used the relatively high-variance estimator (3), upon which we improve. Variable reparameterizations have a long history in the statistics literature, but have only recently found use for efficient *gradient-based* machine learning and inference [4, 13, 19]. Related is also *probabilistic backpropagation* [9], an algorithm for inferring marginal posterior probabilities; however, it requires certain tractabilities in the network making it insuitable for the type of models under consideration in this paper.

As we show here, regularization by dropout [20, 22] can be interpreted as variational inference. DropConnect [21] is similar to dropout, but with binary noise on the weights rather than hidden units. DropConnect thus has a similar interpretation as variational inference, with a uniform prior over the weights, and a mixture of two Dirac peaks as posterior. In [2], *standout* was introduced, a variation of dropout where a binary belief network is learned for producing dropout rates. Recently, [15] proposed another Bayesian perspective on dropout. In recent work [3], a similar reparameterization is described and used for variational inference; their focus is on closed-form approximations of the variational bound, rather than unbiased Monte Carlo estimators. [15] and [7] also investigate a Bayesian perspective on dropout, but focus on the binary variant. [7] reports various encouraging results on the utility of dropout's implied prediction uncertainty.

# 5 Experiments

We compare our method to standard binary dropout and two popular versions of Gaussian dropout, which we'll denote with type A and type B. With Gaussian dropout type A we denote the pre-linear Gaussian dropout from [20]; type B denotes the post-linear Gaussian dropout from [22]. This way, the method names correspond to the matrix names in section 2 (**A** or **B**) where noise is injected. Models were implemented in Theano [5], and optimization was performed using Adam [12] with default hyper-parameters and temporal averaging.

Two types of variational dropout were included. Type A is correlated weight noise as introduced in section 3.2: an adaptive version of Gaussian dropout type A. Variational dropout type B has independent weight uncertainty as introduced in section 3.1, and corresponds to Gaussian dropout type B.

A *de facto* standard benchmark for regularization methods is the task of MNIST hand-written digit classification. We choose the same architecture as [20]: a fully connected neural network with 3 hidden layers and rectified linear units (ReLUs). We follow the dropout hyper-parameter recommendations from these earlier publications, which is a dropout rate of $p = 0.5$ for the hidden layers and $p = 0.2$ for the input layer. We used early stopping with all methods, where the amount of epochs to run was determined based on performance on a validation set.

**Variance.** We start out by empirically comparing the variance of the different available stochastic estimators of the gradient of our variational objective. To do this we train the neural network described above for either 10 epochs (test error 3%) or 100 epochs (test error 1.3%), using variational dropout with independent weight noise. After training, we calculate the gradients for the weights of the top and bottom level of our network on the full training set, and compare against the gradient estimates per batch of $M = 1000$ training examples. Appendix E contains the same analysis for the case of variational dropout with correlated weight noise.

Table 1 shows that the local reparameterization trick yields the lowest variance among all variational dropout estimators for all conditions, although it is still substantially higher compared to not having any dropout regularization. The $1/M$ variance scaling achieved by our estimator is especially important early on in the optimization when it makes the largest difference (compare *weight sample per minibatch* and *weight sample per data point*). The additional variance reduction obtained by our estimator through drawing fewer random numbers (section 2.3) is about a factor of 2, and this remains relatively stable as training progresses (compare *local reparameterization* and *weight sample per data point*).

| stochastic gradient estimator | top layer 10 epochs | top layer 100 epochs | bottom layer 10 epochs | bottom layer 100 epochs |
|---|---|---|---|---|
| local reparameterization (ours) | $7.8 \times 10^3$ | $1.2 \times 10^3$ | $1.9 \times 10^2$ | $1.1 \times 10^2$ |
| weight sample per data point (slow) | $1.4 \times 10^4$ | $2.6 \times 10^3$ | $4.3 \times 10^2$ | $2.5 \times 10^2$ |
| weight sample per minibatch (standard) | $4.9 \times 10^4$ | $4.3 \times 10^3$ | $8.5 \times 10^2$ | $3.3 \times 10^2$ |
| no dropout noise (minimal var.) | $2.8 \times 10^3$ | $5.9 \times 10^1$ | $1.3 \times 10^2$ | $9.0 \times 10^0$ |

Table 1: Average empirical variance of minibatch stochastic gradient estimates (1000 examples) for a fully connected neural network, regularized by variational dropout with independent weight noise.

**Speed.** We compared the regular SGVB estimator, with separate weight samples per datapoint with the efficient estimator based on local reparameterization, in terms of wall-clock time efficiency. With our implementation on a modern GPU, optimization with the naïve estimator took **1635 seconds** per epoch, while the efficient estimator took **7.4 seconds**: an over 200 fold speedup.

**Classification error.** Figure 1 shows test-set classification error for the tested regularization methods, for various choices of number of hidden units. Our adaptive variational versions of Gaussian dropout perform equal or better than their non-adaptive counterparts and standard dropout under all tested conditions. The difference is especially noticable for the smaller networks. In these smaller networks, we observe that variational dropout infers dropout rates that are on average far lower than the dropout rates for larger networks. This adaptivity comes at negligable computational cost.

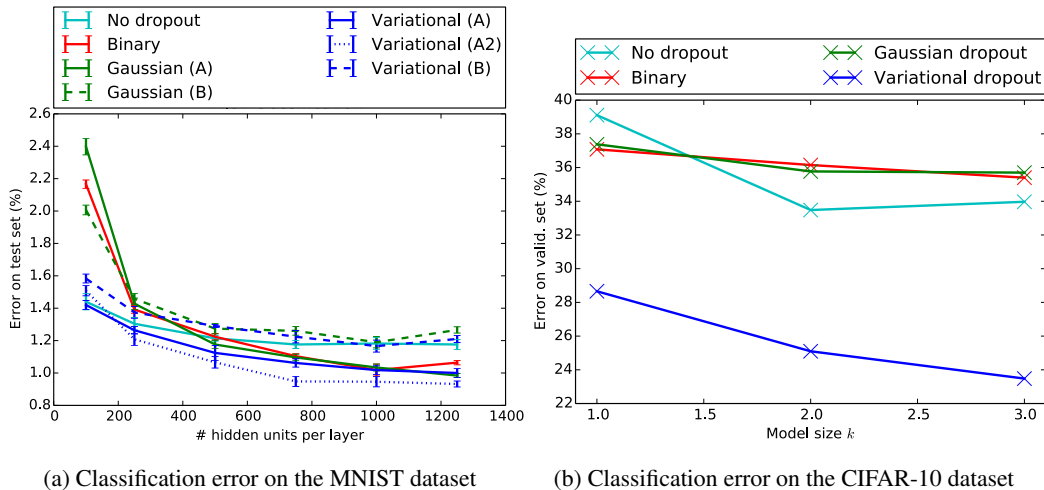

(a) Classification error on the MNIST dataset&emsp;&emsp;(b) Classification error on the CIFAR-10 dataset

Figure 1: Best viewed in color. **(a)** Comparison of various dropout methods, when applied to fully-connected neural networks for classification on the MNIST dataset. Shown is the classification error of networks with 3 hidden layers, averaged over 5 runs. he variational versions of Gaussian dropout perform equal or better than their non-adaptive counterparts; the difference is especially large with smaller models, where regular dropout often results in severe underfitting. **(b)** Comparison of dropout methods when applied to convolutional net a trained on the CIFAR-10 dataset, for different settings of network size $k$. The network has two convolutional layers with each $32k$ and $64k$ feature maps, respectively, each with stride 2 and followed by a softplus nonlinearity. This is followed by two fully connected layers with each $128k$ hidden units.

We found that slightly downscaling the KL divergence part of the variational objective can be beneficial. *Variational (A2)* in figure 1 denotes performance of type A variational dropout but with a KL-divergence downscaled with a factor of 3; this small modification seems to prevent underfitting, and beats all other dropout methods in the tested models.

## 6&ensp;&ensp;Conclusion

Efficiency of posterior inference using stochastic gradient-based variational Bayes (SGVB) can often be significantly improved through a *local reparameterization* where global parameter uncertainty is translated into local uncertainty per datapoint. By injecting noise locally, instead of globally at the model parameters, we obtain an efficient estimator that has low computational complexity, can be trivially parallelized and has low variance. We show how dropout is a special case of SGVB with local reparameterization, and suggest *variational dropout*, a straightforward extension of regular dropout where optimal dropout rates are inferred from the data, rather than fixed in advance. We report encouraging empirical results.

### Acknowledgments

We thank the reviewers and Yarin Gal for valuable feedback. Diederik Kingma is supported by the Google European Fellowship in Deep Learning, Max Welling is supported by research grants from Google and Facebook, and the NWO project in Natural AI (NAI.14.108).

## Footnotes

[1]Note that the described method is not limited to classification or regression and is straightforward to apply to other modeling settings like unsupervised models and temporal models.

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
