[Reviews · NeurIPS 2015]

Submitted by Assigned_Reviewer_1

This paper is very well-written and clear.

The intro provides a good overview of the sub-field, the technical exposition is clear, and the example is also well-motivated and clear.

I would have liked to see a visualization of the per-parameter optimized dropout rates!
Summary: This paper proposes improved gradient estimators for SVI, and also makes a connection to dropout.

This new perspective allows them to generalize dropout and learn the dropout parameters using the marginal likelihood bound.

Submitted by Assigned_Reviewer_2

*** This is as light review ***

The authors introduce a 'local reparameterization trick', to reduce variance in stochastic gradient estimators, and then relate this to Gaussian dropout, showing that dropout maximizes a variational bound.

Quality -- A nice paper, excellent writing brings several ideas together. I enjoyed the note on floating point numbers!

Originality -- Builds on several contemporary ideas, making connections between them.

Clarity -- As clear as one could expect given the page limit. Appropriate use of appendices.

Typo at line 100: did you mean f(.) is a differentiable function?
Summary: Nice paper connecting variational inference and dropout.

Submitted by Assigned_Reviewer_3

**Updated review after authors' responses***

The main novelty/usefulness of the paper is related to the points: *usage of separate sample of weights per data-points and the local reparametrization trick". Even these ideas are not so novel. For example, the local reparametrization trick is something that we use all the time when we do Variational Bayes (VB) (say in a logistic regression model) and transform high-dimensional integrals into one-dimensional integrals under a Gaussian approximate posterior. For example, if you have a likelihood of the form

\prod_{i=1}^n \sigma(w^T x_i)

and apply VB with q(w|mu,Sigma), then you end up with a sum of expectations of the form

\sum_{i=1}^n q(w|mu,Sigma) \log \sigma(w^T x_i) d w

and then the local reparametrization trick is applied to transform each separate (initially high-dimensional integral over the vector w) into a 1-D integral over the univariate standard normal. The authors essentially use this separately for each activation unit and apply stochastic approximation instead of integration. Having said that, I must admit that as far as the stochastic variational inference algorithms are concerned and the related research community (born a couple of years ago!) the use of this local reparametrization trick, as far as I know, is novel and people should know about it because it is useful.

In contrast, I believe that VB-Dropout message of the paper is very limited. This is because the paper simply explains existent Dropout algorithms by VB, but it does not go any step further to tells us reliably something new, i.e. whether such a connection is useful for either VB or Dropout.

Specifically. what are really the VB-Dropout connection messages that the paper tries to convey? Are they that

1) in order the get the benefits of Dropout we should be just applying VB with a factorized Gaussian distribution on the weights? Should we parametrize the variational distribution as N(w|theta, a*theta^2)? If the latter is true and we use a separate theta and alpha for each different weight (the authors claimed in their feedback that one of their methods referred to as "Gaussian dropout type B" is precisely this approach), then this the same with the regular parametrization N(w|mu,sigma^2) since both variational distributions have the same approximation capacity. If the last is true, then there is not really any point of using N(w|theta, a*theta^2).

2) or the message is that we should be doing VB with a factorized variational Gaussian distribution of the form N(w|theta, alpha*theta^2), but where we additionally place some restrictions on the values of alphas plus this very weird (to my opinion!) improper log-uniform prior on the weights that lead to Dropout? Is this the take-home message?

3) or that based on VB we can learn the Dropout rate parameters, i.e. the alphas? According to Section 3.4 during learning, these alphas are restricted to be smaller that 1!! So what is the message here? Is is that with VB we can sort of learn these alphas, but we are not quite there yet? This is completely unclear.

I don't think the paper gives us a clear conclusion as far as the potential usefulness of the VB-Dropout connection. To give further examples (answered questions): Should we be using the improper prior or the Gaussian prior on the weights? Or If we use the Gaussian prior it is going to really damage the prediction accuracy?

In terms of the question "Should we be using just VB with factorized unrestricted Gaussian to get the benefits of Dropout?" I asked this in my initial review whether this method was included in the experiments. The authors replied the following: **What we call "Gaussian dropout type B" is the unrestricted variational approximation: that is, the factorised Gaussian posterior approximation with independent variances per parameter. This method is included in both experiments. This approximation (like any factorised posterior) can still be interpreted as being equivalent to dropout; see Appendix C. We'll clarify this in the text.**

But I am not sure if they are saying the truth because in the section 3.4 they write: **Although performance is quite robust to different specifications of alpha, we do find it beneficial to set an upper bound for the alpha parameters during training. Using our Gaussian posterior approximations proposed above, we allow for a maximum value of alpha = 1, i.e. we maximize the posterior variance at the square of the posterior mean, which corresponds to a dropout rate of 0.5. This way, we avoid bad local optima of the variational objective that cause the noise to grow very large.** => So was it unrestricted or restricted?

So overall I think that the local reparametrization aspect of the paper applied to SVI is useful and this a good point about the paper. But, in contrast, I have found the VB-Dropout part of the paper with no any clear message or conclusion.

***Review before authors' responses***

The paper proposes a stochastic gradient variational Bayes (SGVB) framework for neural networks that reduces the variance of the stochastic gradient in the initial SGVB scheme (see Kingma and Welling; Rezende et al)). This is achieved by sampling from the noise distribution (standard normal) in a per-datapoint basis rather than by sampling globally for the full data minibatch. The part of the paper that describes this idea, the so called "local reparametrization trick", lacks of clarity and the writing is extremely confusing (even now I am still not sure whether there is a fatal technical error or not in the derivation in Section 2). More precisely, the whole idea is to reparametrize the variational distribution, when approximating an intractable expectation under this distribution, not based on the parameters W (the standard SGVB way) but based on the neuron activations B (ie the linear transformations of the actual parameters W that are passed as inputs in the subsequent layer). The derivation of the Gaussian distribution of these neuron activations, according to eq. (6), is confusing since according to (6) the Gaussian over B is not fully factorized (as the authors claim) but each column of B, say b_i is fully dependent with covariance matrix equal to A*diag(Sigma_i)*A^T. The authors need to carefully explain the derivation in eq (6) and be very clear about whether there is a separate W for each data point or not (so that the reader should not make guesses about what is going on). Despite this confusing writing the local reparametrization trick seems to reduce variance and speed up the SGVB procedure for neural nets.

The second contribution of the paper is that it tries to explain dropout regularization as maximization of a certain variational lower bound where the Gaussian variational distribution

is constrained accordingly in order stochastic maximization of the bound to lead to some variant of Gaussian dropout. Specifically, the authors manage to give an alternative explanation of the Wang and Manning (2013) dropout procedure based on their local reparametrization trick. This has the potential benefit that the dropout rate parameters could be learnt by maximizing the lower bound. The experiments show some small improvement in MNIST and CIFAR-10, but they don't look so conclusive and the authors do not give much information. How did you run the method "No dropout"? Was it your improved version of SGVB procedure but with using an unrestricted (ie. not restricted to give rise to dropout) variational distribution? If not then you MUST add this method in the comparison. In CIFAR you must include in the comparison the method "No dropout" and "A2" that you have considered for MNIST.

Summary: Speed up of SGVB for neural nets. The paper is badly written and the experiments are preliminary.

Submitted by Assigned_Reviewer_4

This paper presents important improvements for doubly stochastic variational inference (where variance in the variational lower bound and its gradients results from both the use of mini-batches and Monte-Carlo estimates of the expected log-likelihood.) The "local reparemeterization trick" introduced in this paper achieves great computational savings together with lower variance in the Monte-Carlo estimator of the lower bound. The paper also shows that dropout is equivalent to doubly stochastic variational inference with the local reparemeterization trick and a particular prior on the weights. A very interesting feature of the presented method is that the dropout rate (or rates) can be optimized simultaneously with the rest of parameters.

The paper is very clearly written with an emphasis on transmitting the key technical ideas rather than providing an over-rigorous exposition.

The work is original and very interesting. A similar connection between dropout and variational inference has been recently shown by Gal and Ghahramani. In particular, they show that dropout in a multi-layer perceptron is equivalent to maximizing the variational lower bound of a deep Gaussian process model.

Minor comments:

What is g(.) in section 2.1?

Colors for methods A and B are inverted between the two panels of Figure 1.
Summary: This paper presents very interesting improvements and insights on the important problem of doubly stochastic variational inference.

Submitted by Assigned_Reviewer_5

** Updated review after authors' response **

I spent some time trying to get a broader view of the work since I wrote my review. To be honest, my view now is that the paper is very well written, but with almost no novel contribution.

The main idea is "sample a new random variable realisation for each data point in the mini-batch". I have been using this technique for most of my applications working with SVI - it is fairly obvious (although the theoretical variance analysis is novel as far as I know). It does have some computational limitations that the authors avoid by the re-parametrisation trick - which I'm afraid is *very* problem specific. I can't easily see how this trick could be used for models other than MLPs. The author's answer "We show that dropout can be interpreted as being as a result of, rather than being equal to, a local reparameterization of weight uncertainty" suggests that the authors claim dropout to be a special case of the technique; this would have been fine if other, non-trivial, models were used as well. This is my main concern, that the paper would have been *a lot stronger* (thus beneficial to the community) if the authors were to suggest and assess this with general models (rather than models that make use of it anyway, see next). The authors do mention in the rebuttal that this should be applicable to "a larger family of graphical models with plates" which I think is what we would have been most interested in.

My second major concern is that the models used in the assessment use this trick anyway. Even though the authors claim in the rebuttal that earlier methods do not use the technique, I am certain that all implementations of dropout (with all its variants) sample new random variable realisations for each point in the mini-batch. I spent a lot of time studying different open-source implementations and, even though this is sometimes overlooked in the literature, the actual implementation would do this. The dropout journal paper mentions this explicitly, and the popular Caffe open source framework implements this as well. Another example is the increasingly popular Keras framework which implements Gaussian multiplicative noise as well as other noise layers, all of which sample new realisations for each data point (https://github.com/fchollet/keras/blob/master/keras/layers/noise.py). I would note that fast dropout operates differently and matches the first two moments of the input. That's why I find the experiments not satisfying. The dropout MLP models are presumably just the vanilla models used in the wild. The main improvement is the optimisation of the dropout parameter. But this was already done before in [A Bayesian Encourages Dropout, Maeda] and can't be counted as a novel contribution (for that matter Shin-ichi also discusses a Bayesian interpretation of dropout).

The novel contributions of the paper seem to be

1. "sample separate random variable realisations per data-point in stochastic variational inference to reduce the estimator variance" (which I think is important, but somewhat known) and

2. an explanation for dropout's low variance optimisation objective (although it might be just my interpretation as I believe this is not mentioned explicitly as an insight for dropout from the analysis).

After a long deliberation I have decided to reduce my score slightly as the paper does not communicate these contributions well in its current form. Even though the paper is written well (in the sense that it is very pleasant to read), both the authors and the community would have benefited greatly from re-structuring the paper to present these more focused ideas.

** Review before authors' response **

The main results of the work are very interesting. The analysis (both theoretical and experimental) of the estimator variance of both fast dropout and multiplicative Gaussian noise dropout is important and enlightening.

Major criticism: * The main contribution of the paper is linking multiplicative Gaussian noise dropout and fast dropout to variance reduction techniques in a Bayesian setting, as is mentioned explicitly in the introduction ("in this paper we show that a relationship between dropout and Bayesian inference can be extended and exploited [...]").

This is stretched rather thinly as a new technique for gradient based variational inference ("we explore an as yet unexploited trick for drastically improving the efficiency of stochastic [...]") even though the "trick" mentioned is known and has been used for quite some time: both multiplicative Gaussian noise dropout and fast dropout perform this, and this is not new by itself. I would have expected to see the authors use this technique for new models in the experiments section rather than presenting this as a new development and then assessing this with models that *already* use this technique.

* I'm confused about the generality of the method suggested. In line 107 it is mentioned that the method can be applied to any probabilistic model, which is true for sampling a new random variable for each data point (as is often done anyway). But the computationally efficient form in line 171-180 is only applicable to neural networks. Furthermore, experiments are only done for neural networks, with all models compared simply being multiplicative Gaussian noise dropout and fast dropout with the only novelty being the tuning of the dropout parameter. * section 3.3: the use of a log-uniform prior seems quite forced (mostly to get rid of the KL divergence term as a function of theta). Especially given that dropout is often used with L2 regularisation in many models, which would add additional unaccounted-for theta terms to the objective function. Furthermore, the fact that the authors have to limit the value of alpha (section 3.4) shows that the effects of the regularisation are not sufficient.

Minor criticism: * The citation format is incorrect. * Lines 185-186: It is not clear whether by a "local re-parameterisation trick" the authors mean sampling a separate random variable for each data point in the mini-batch, or specifically sampling a separate random variable for each *unit* in a network. * Lines 57-59: this is wrong; [1] has shown identical results to dropout in a Bayesian setting. * Line 144: why is cov positive on average?

* Line 404: why wasn't the dropout probability p scaled down for the smaller networks? what are the alpha values used with fast dropout and multiplicative Gaussian noise dropout? * Lines 410-413: why wasn't Variational (A2) assessed for CIFAR-10 as well?

I hope the authors will find the following minor comments helpful: * Many citations have proper names in lower case (such as fisher, bayesian, langevin, etc). You can fix this by surrounding these words with curly brackets in the bib file ("... fisher ..." -> "... {F}isher ..."). * Line 61: "data(Kingma" * Line 100: g() should be f() * Line 112: "are correct on average" is quite informal * Line 132: why does the equation contain the term L_D(phi)^2/N^2 instead of E(L_i)^2? * Line 215: "with a M x K of..." should be "with an M x K matrix of..." * Line 216: "by adding noise ... the weights are less likely to overfit" - why?

* Line 233: Wang and Manning (2013)'s derivation uses Bernoullis on the inputs to the product, and the marginal was *approximately* Gaussian. Presumably the reason for the good performance for both dropout and fast dropout is that both Gaussians and Bernoullis had the low variance discussed in this paper (compared to distributions over the weights); fast dropout simply approximated the output of the Bernoulli times weights. * Line 239: "activation noise" is not defined * Line 255-257: a bit confusing * Line 262: "... for which the noise enters this way" is not clear * Lines 290-292: can be moved to appendix * Line 303: "performance is quite robust to different specifications of alpha" - presumably because the model will simply adapt the weights to match the moments of the full posterior? * Line 318: \ref{} should be \eqref{} for equation 3 * Line 330: "similar to our local reparameterization trick" - not so much; Bayer simply matches the first two moments of the input distribution in a variational setting, equivalent to fast dropout with generalised noise processes. * Section 4: you might consider adding references to [2], [3] * Line 362: "lower" should be "lowest" * Lines 371-373: it is not clear that the three models correspond to random variable samples per unit / samples per weight / samples per batch (or data point?)

References:

[1] Weight Uncertainty in Neural Network Charles Blundell, Julien Cornebise, Koray Kavukcuoglu, Daan Wierstra, ICML, 2015 http://jmlr.org/proceedings/papers/v37/blundell15

[2] A Bayesian Encourages Dropout Shin-ichi Maeda Under review as a conference paper at ICLR 2015 http://arxiv.org/pdf/1412.7003.pdf

[3] Dropout As A Bayesian Approximation: Insights And Applications Yarin Gal, Zoubin Ghahramani Deep Learning Workshop, ICML, 2015

https://sites.google.com/site/deeplearning2015/33.pdf?attredirects=0

Summary: The authors show that sampling a new random variable for each data point in a mini-batch results in a low variance estimator. They interpret multiplicative Gaussian noise dropout and fast dropout as variational methods and propose generalisations, optimising over the dropout parameter. The paper is written well. ** Please see updated review after authors' response **

Author Feedback
Author rebuttal: Reviewers, thank you for your time and very informative and useful feedback. While the reviews were quite positive overall, we will here focus on the main points of critique.

=== Assigned_Reviewer_1 ===

Remark:
"The derivation of the Gaussian distribution of these neuron activations, according to eq. (6), is confusing [...]."

Answer:
You are correct that the posterior distribution of B is not factorised over the rows (the training examples), but the required expectations for evaluating the variational objective only depend on the marginal distributions of this posterior for each separate training example, not on any interactions between training examples. It is therefore correct to sample a separate weight matrix W from the posterior for each row of A (i.e. each data point), as stated in the paragraph preceding equation 6. We will make this more explicit.

The elements within each row of B (i.e. the columns) are independent under the posterior distribution on B, since they are each functions of different columns of W (and our posterior is factorised over these columns).

Question:
"How did you run the method "No dropout"? Was it your improved version of SGVB procedure but with using an unrestricted [...] variational distribution?"

Answer:
The "no dropout" method is without any regularisation (except early stopping). What we call "Gaussian dropout type B" is the unrestricted variational approximation: that is, the factorised Gaussian posterior approximation with independent variances per parameter. This method is included in both experiments. This approximation (like any factorised posterior) can still be interpreted as being equivalent to dropout; see Appendix C. We'll clarify this in the text.

=== Assigned_Reviewer_3 ===

Major criticism 1:
"* [the local reparameterization trick] is stretched rather thinly as a new technique [...]. I would have expected to see the authors use this technique for new models [...]."

Answer:
Earlier methods (including multiplicative Gaussian noise dropout and fast dropout) do *not* use the technique. We show that dropout can be interpreted as being as a result of, rather than being equal to, a local reparameterization of weight uncertainty. We feel that the application of local reparameterization to stochastic gradient VB is non-trivial, as earlier works in this field (i.e. the recent "Weight Uncertainty in Neural Networks" paper you reference) did not use this technique, even though it would have been highly beneficial.
Also, since we *infer* the dropout rates per unit or weight, our models are different, rather than equal to earlier models.

Major criticism 2:
"* [...] In line 107 it is mentioned that the method can be applied to any probabilistic model, [but] is only applicable to neural networks."

Answer:
We only write that it applies to any probabilistic model *setting* (generative, temporal, etc.). We will improve the wording to avoid any confusion. The method is in fact applicable to a larger family of graphical models with plates.

Major criticism 3:
"* section 3.3: the use of a log-uniform prior seems quite forced (mostly to get rid of the KL divergence term as a function of theta). [...]. Furthermore, the fact that the authors have to limit the value of alpha (section 3.4) shows that the effects of the regularisation are not sufficient."

Answer:
As explained in appendix A, the log-uniform prior corresponds to the prior for which the floating-point number format is approximately optimal.
Your last point about limiting the value of alpha is not valid. Upper bounding alpha (as we do) limits the dropout rate, potentially only weakening the regularisation (not strengthening it).

Minor criticism:
"* Line 144: why is cov positive on average?"

Answer: The covariance term referred to is the covariance between the likelihood terms L(x_i, w), L(x_j, w) for different data points x_i and x_j, taken over the approximate posterior distribution of the weights w. Assuming that x_i and x_j are drawn i.i.d. from the same distribution, we have that E_x_i[ L(x_i, w) ] = E_x_j[ L(x_j, w) ] for any value of w. Assuming that the likelihood terms are non-constant over the domain of our posterior distribution on w, this means that their expected covariance (expectation with respect to x_i, x_j, covariance with respect to w) is positive. We will make this explicit in the text.

Minor criticism:
"* Line 404: why wasn't the dropout probability p scaled down for the smaller networks? what are the alpha values used with fast dropout and multiplicative Gaussian noise dropout?"

Anwer:
Searching for optimal dropout rates for each network size would be computationally very expensive. We followed the rates recommended in previous dropout publications, and used the same values for the upper bounds in variational dropout.

We'll add the missing comparisons for CIFAR in the final version of the paper.